# Unravelling the Temporal and Chemical Evolution of a Mineralizing Fluid in Karst-Hosted Deposits: A Record from Goethite in the High Atlas Foreland (Morocco)

Michèle Verhaert [1,2], Cécile Gautheron [2], Augustin Dekoninck [1,*], Torsten Vennemann [3], Rosella Pinna-Jamme [2], Abdellah Mouttaqi [4] and Johan Yans [1]

1. Institute of Life-Earth-Sciences, Department of Geology, University of Namur, Rue de Bruxelles 61, 5000 Namur, Belgium
2. Geosciences Paris Saclay (GEOPS), Université Paris Saclay, 91404 Orsay, France
3. Institute of Earth Surface Dynamics, University of Lausanne, Géopolis, CH-1015 Lausanne, Switzerland
4. Office National des Hydrocarbures et des Mines, Avenue Moulay Hassan 5, BP 99, Rabat 10000, Morocco
* Correspondence: augustin.dekoninck@unamur.be

**Abstract:** Timing and duration of ore deposit formation are crucial to understanding the mineralization process. To address this, the geochronological (U-Th)/He method, geochemical and H- and O-isotope compositions of pure goethite formed in the Imini karst-hosted Mn district (High Atlas, Morocco) were examined in detail. Two main generations of cavity-filling and fracture-filling goethite are identified, and both precipitated prior to the massive Mn oxide ore. The $\delta D$ and $\delta^{18}O$ values reveal that the mineralizing fluid of cavity and fracture-filling goethite is meteoric-derived but enriched in $^{18}O$ due to fluid–rock interactions with the host rock dolostone or mixing with $O_2$-rich surface water resident in an open karst system. The cavity-filling goethite precipitated between 95 to 80 Ma, whereas fracture-filling goethite formed between 80 to 50 Ma. Ore deposition occurred discontinuously during the early Atlas doming associated with one or more early compressional events in the Atlas tectonism. The increase in $\delta D$ values and depletion in U content result from a change in the mineralizing fluid within the karst system. At about 50 Ma, the fluid is notably enriched in U, Cu and trace metals.

**Keywords:** karst-hosted; goethite; (U-Th)/He dating; stable isotope; $\delta D$; $\delta^{18}O$; manganese; Imini; anti-atlas; Morocco

## 1. Introduction

Timing and duration of fluid circulation and associated ore deposition are often difficult to constrain, notably as few ore minerals can be dated. Age constraints on supergene and subsurface ore deposits come mostly from the use of K-Ar and $^{40}Ar/^{39}Ar$ methods on K-Mn oxides [1] and using the (U-Th)/He method on goethite (GHe) [2,3]. The latter method has received increased attention over the years due to its ease of use and the ubiquitous presence of goethite at the Earth's surface. The method has been applied to dating ore deposit formation [2,4,5], supergene weathering of porphyry [6] and determining the age of lateritic iron duricrust [7–10]. In addition, the study of U and Th content in goethite reveals that goethite precipitating from circulating fluid in the water table contains U at ppm levels and almost no Th due to the solubility difference between those two elements [9]. Ultimately, this allows the distinction between autochthonous from allochthonous goethite precipitation.

We combined petrological, mineralogical, geochemical, H- and O-isotopes and GHe geochronological data on pure goethite from the Imini district (High Atlas, Morocco) to discuss the age and evolution of the mineralization through time. The choice of the Imini district was based on: (i) the existence of previous detailed studies dealing with ore genesis

of the main Mn ore [11–14]; (ii) the provision of large amounts of massive and fairly pure botryoidal goethite and (iii) the presence of K-bearing Mn oxides and their potential $^{40}Ar/^{39}Ar$ dates.

## 2. Geological Setting

The High Atlas of Morocco results from intraplate Cenozoic compression along a belt that has experienced several compressional and erosional phases (Figure 1a). The first one is poorly defined and occurred during the late Cretaceous [15], whereas the subsequent three Cenozoic phases building the High Atlas are better documented [16]. One particularity of the High Atlas range is that the present-day foreland basins are poorly deformed but contain numerous ore deposits [17,18]. Some of them are affected by later weathering processes (e.g., [19,20]), in particular the southern foreland basin host Mn ore deposits in Cretaceous rocks [14]. In the Ouarzazate basin, the Imini ore deposit forms lenses aligned along a WSW-ENE direction over 25 km, hosted in a ~10 m-thick Cenomanian-Turonian dolostone, extending to the Tasdremt ore deposit to the west in the Souss basin (Figure 1b,c). The Imini Mn resources are estimated to be 1,500,000 metric tons [21] with grades higher than 70 wt.% MnO [11].

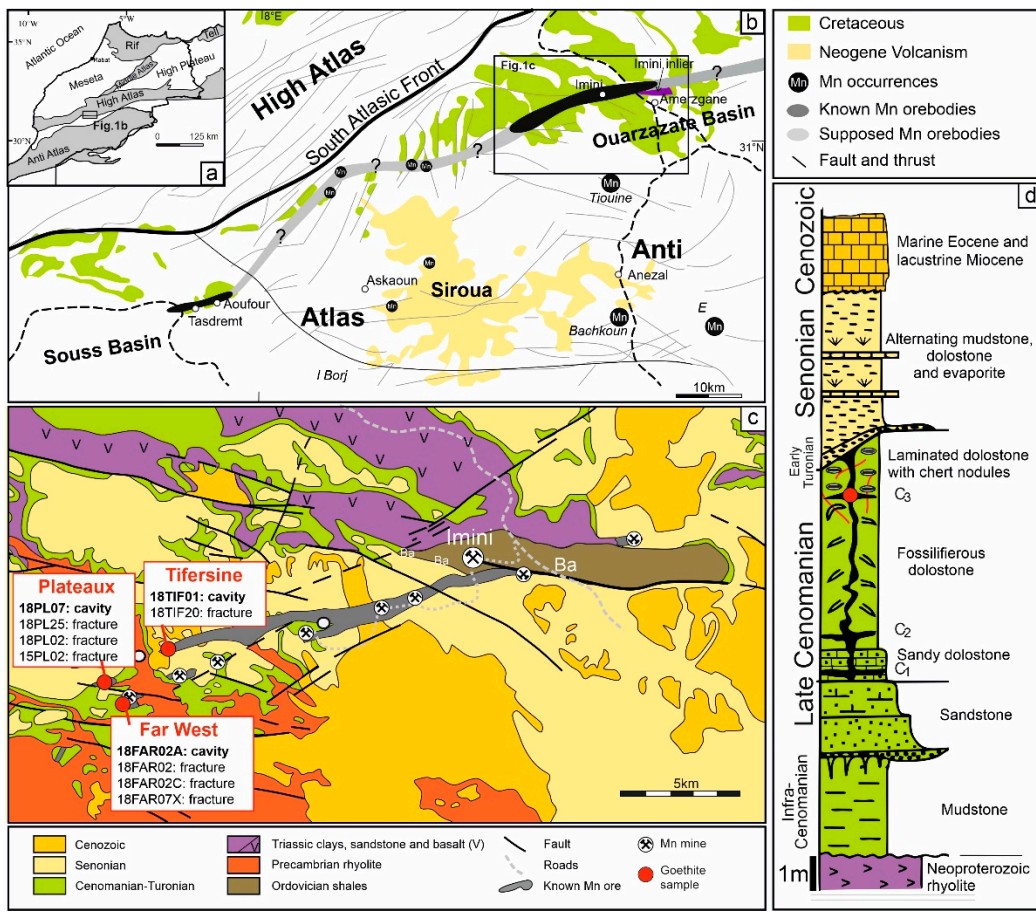

**Figure 1.** (**a**) Simplified structural map of Morocco. (**b**) Geological map of the Cretaceous sediments, Neogene volcanism and Mn occurrences [14]. (**c**) Geological map of the Imini district showing the location of Far West, Tifersine and Plateaux deposits [12]. (**d**) Stratigraphic log of the western part of the district [13]. Cross-section is presented in [11].

Early diagenetic dolomitization occurred rapidly after the deposition of the late Cenomanian to early Turonian carbonate succession [13,22]. Karst cavities filled by Senonian (Upper Santonian according to [23]) sediments attest that karstification occurred after the relative sea level decrease and thus early after dolomitization of the Cenomanian-Turonian dolostone (Figure 1c) [13]. Dekoninck et al. [12] have additionally demonstrated that the

Mn ore formation occurred with the dissolution of the host rock dolostone. This is observed through infills of Mn oxides in pseudo-breccia. The Mn ore is subdivided into three 1 to 2.5 m-thick stratabound orebodies defined as C1, C2 and C3 [11–13]: the lowermost C1 and C2 are predominantly composed of pyrolusite, and the uppermost C3 is composed of coronadite group minerals (Figure 1d). Pyrolusite and coronadite group minerals are present in each of the three stratabound levels. These Mn oxides precipitated during an epigenetic stage (1) by replacement of the dolomite fabric, and a colloform stage (2) in open voids (Figure 2). Fe oxides and hydroxides occur as accessory minerals in the widespread Mn ore in the northern front of the Imini Mn ore deposits [24,25]. Fe oxides are absent elsewhere in the district and occur early in the paragenetic sequence (Figure 2). They occur in the upper C3 orebody only (Figure 1c).

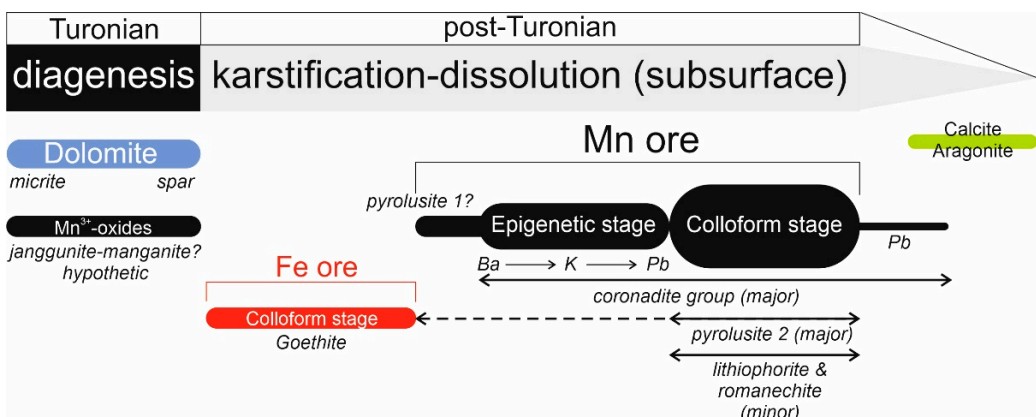

**Figure 2.** Simplified and modified paragenetic sequence. The position of the Fe mineralization prior to the Mn oxide is illustrated. Adapted from [12].

## 3. Materials and Methods

### 3.1. Sampling Strategy

The most representative sampling sites of botryoidal goethite are located in the western sector (Figures 1c and 3), where 19, 20 and 34 samples of iron mineralization were collected in Far West, Tifersine and Plateaux, respectively. Coronadite group minerals, goethite and minor hematite are concentrated in the uppermost C3 ore body [12,13]. Botryoidal goethite is the most common iron phase at Imini, and poorly crystallized hematite and goethite are found in C1 and C2 orebodies. Goethite does not replace the dolomitic host rock (epigenetic stage) and is not substituted by later Mn oxides as illustrated in Figure 3.

Hand-sized samples were collected in ore dumps, veins/fractures in outcrops (Figure 3A) and in cavities on galleries (Figure 3B). Different types of iron mineralization were selected in the field, and eight pure botryoidal goethite samples were selected for (U-Th)/He geochronological and δD-δ$^{18}$O isotopic measurement (Table S1). Botryoidal goethite was estimated to be the purest and best-crystallized material and was further examined (Figure 4).

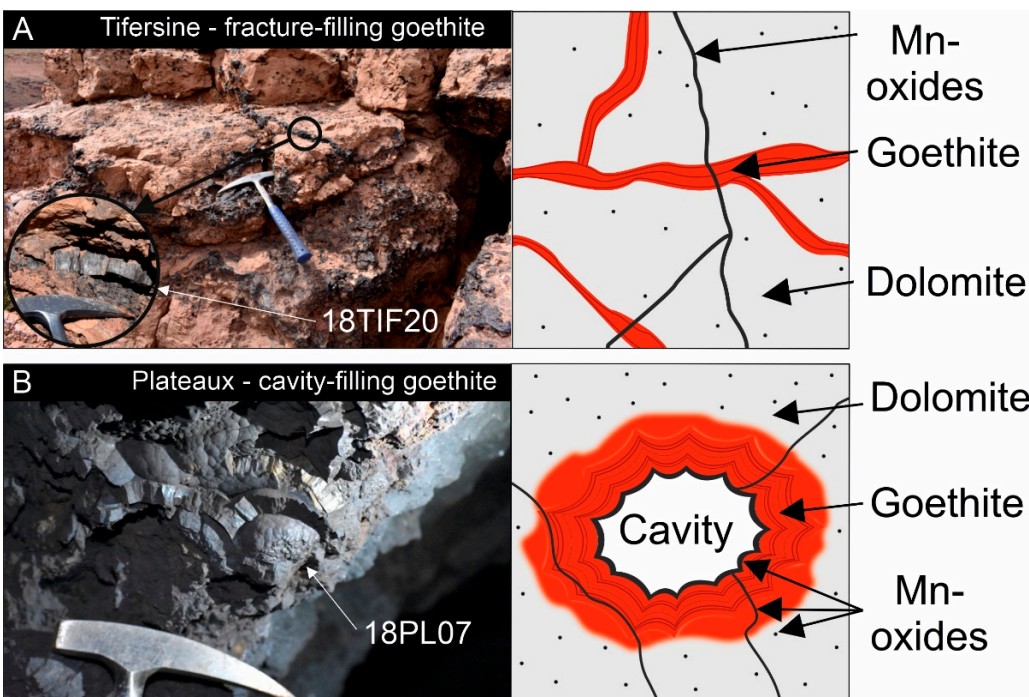

**Figure 3.** Style of the Fe mineralization and schematic representations. (**A**) Botryoidal goethite filling fractures and forming thin veins at Tifersine site (sample 18TIF20). (**B**) Large botryoidal goethite filling cavities at the Plateaux site (sample 18PL07).

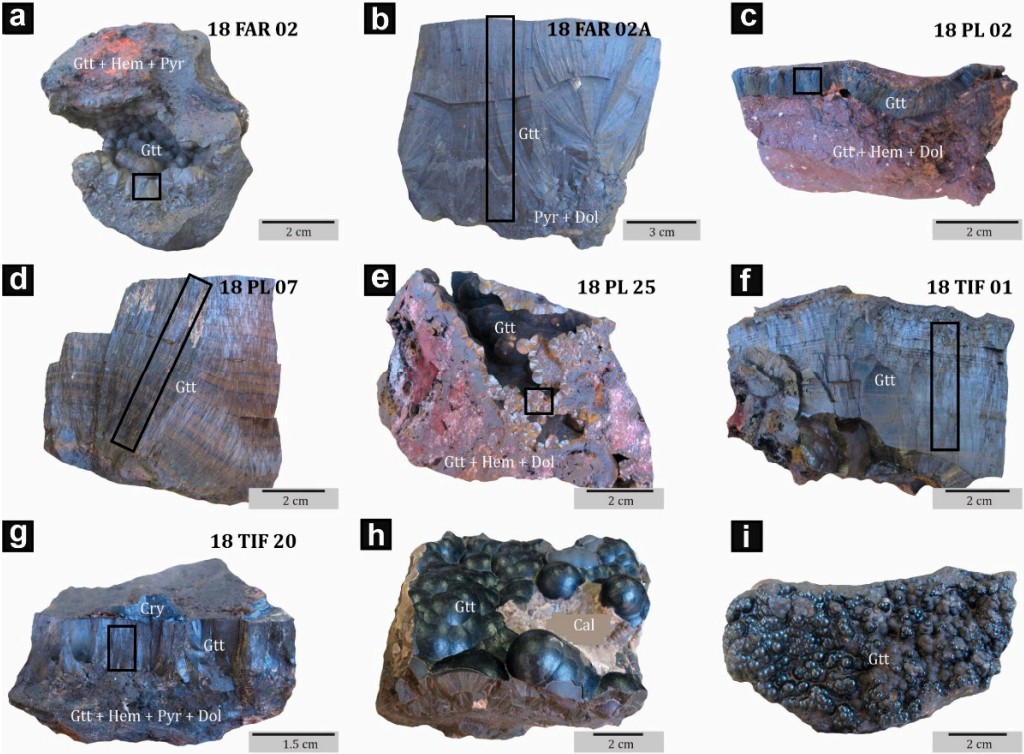

**Figure 4.** Pictures of botryoidal goethite samples. The distinction between goethite filling thin fractures (**a,c,e,g**) and large goethite filling cavities (**b,d,f,h,i**) is shown. The black frame indicates the area sampled for (U-Th)/He dating and isotopic analyses. Goethite (**h,i**) are exposition specimens for which no (U-Th)/He dating or isotopic analyses were carried out. Gtt = goethite; Hem = hematite; Cry = cryptomelane; Cal = calcite.

### 3.2. Petrological and Mineralogical Characterization

X-ray diffraction patterns were measured on nine samples from Far West, five from Tifersine and eight from Plateaux (Table S1). A Philips Analytical X-ray diffractometer (Malvern Panalytical, Spectris, Egham, UK) with a Cu anticathode was used to identify the major mineral phases (University of Namur, Namur, Belgium). Eight, three and seven polished sections from Far West, Tifersine and Plateaux, respectively, were studied using a Zeiss Axiophot reflection microscope (Carl Zeiss AG, Oberkochen, Germany) and with a Jeol JSM-7500F scanning electron microscope (SEM) (JEOL, Tokyo, Japan) coupled to an energy dispersive electron spectrometer (EDS) (Table S1) at the University of Namur (Namur, Belgium). From each site, we selected colloform samples containing numerous growth bands. These samples were examined with a Zeiss Sigma 300 FEG scanning electron microscope (Carl Zeiss AG, Oberkochen, Germany) equipped with two Bruker EDS detectors (Bruker Corporation, Billerica, MA, USA) at the University of Liège (Liège, Belgium).

### 3.3. Geochemical Characterization

Geochemical compositions were measured on four samples from Far West, three from Tifersine and five from Plateaux in the Activation Laboratories of Ancaster (Ancaster, ON, Canada) (Table S1). Rare earth and major elements were analyzed with fusion mass spectrometry (FUS-MS; Perkin Elmer Sciex Elan 9000 ICP-MS; Sciex AB, Marsiling, Singapore) and fusion inductively coupled plasma optical emission spectrometry (FUS-ICP; Varian Vista 735 ICP; Agilent, Santa Clara, CA, USA), respectively. Trace elements were quantified by FUS-MS, with the exception of V, Sr, Zr, Ba, Sc and Be, which were determined by FUS-ICP. FeO was quantified by titration. Results are reported in Tables S2–S4.

### 3.4. Stable Isotope Geochemistry

The oxygen ($\delta^{18}O$) and hydrogen ($\delta D$) isotopic compositions were determined for two samples of botryoidal goethite of Far West (18FAR02 and 18FAR02A), two of Tifersine (18TIF01 and 18TIF20) and four of Plateaux (15PL02, 18PL02B, 18PL07 and 18PL25) at the Institute of Earth Surface Dynamics of the University of Lausanne (Switzerland) (Table S1). Hydrogen and oxygen isotopic compositions were measured with a Finnigan MAT253 gas source mass spectrometer (Thermo Fisher Scientific, Waltham, MA, USA) and are reported in per mil (‰) in the typical δ-notation relative to the Vienna Standard Mean Ocean Water (VSMOW) standard. A method adapted from [26] was used for the oxygen isotope measurements using a $CO_2$-laser fluorination line. Between 2 and 3 mg of goethite, crushed from individual grains, was loaded onto a small platinum sample holder. After several cycles of pre-fluorination of the chamber, samples were heated in the presence of 50 mbar of $F_2$ with a $CO_2$-laser to yield $O_2$. After purification of the gas, the extracted $O_2$ was introduced into the inlet of the mass spectrometer. Replicate oxygen isotope analyses of the LS-1 quartz standard (in-house reference material of Lausanne with an accepted $\delta^{18}O$ value of 18.1‰) gave an average value of $18.10 \pm 0.24$‰ for $\delta^{18}O$. The hydrogen isotope composition and water content of goethite were determined with a zero blank auto-sampler and a High-Temperature Conversion Elemental Analyzer (TC-EA) (Thermo Fisher Scientific, Waltham, MA, USA) according to the method of Bauer and Vennemann (2014) [27] and using 2 and 3 mg of goethite. In-house reference materials of biotite (G1; $\delta D = -62 \pm 0.8$‰) and kaolinite (K-17; $\delta D = -125 \pm 0.9$‰) were used to calibrate the measured isotopic compositions [27]. All the measurements of samples and standards were replicated, but only their average values are reported in Table S5.

### 3.5. Goethite (U-Th)/He Dating

(U-Th)/He data were determined for eight pure and well crystallized cm-sized botryoidal goethite samples (Figure 4b,d,f) at the GEOPS laboratory of Paris Saclay University (France). As the method is based on He production and accumulation within the crystal structure due to alpha-decay of U, Th and Sm incorporated in minerals [28,29], the samples

were firstly prepared and selected for He analysis, followed by digestion and U, Th and Sm analysis. Goethite filling thin fractures was carefully crushed, producing ~500 μm long fragments (needles) that were cleaned with distilled water in an ultrasonic bath and later rinsed with pure ethanol. The clean fragments were selected by handpicking using a binocular microscope on the basis of their homogeneity in texture and color, coherence and low porosity. Five aliquots were selected for each sample located into the thin veins (18FAR02, 18TIF20, 15PL02, 18PL02B, 18PL25; Figure 4). For large-size goethite samples tailored to sawed smooth surfaces (18FAR02A, 18TIF01 and 18PL07), fragments from three different layers of crystallization were extracted by means of a drill tool (Dremel with titanium tips. Each section of the very thin growth bands represents a single event of goethite precipitation: the grains dated by the (U-Th)/He method may contain not just one but several successive generations of goethite, as observed by Heim et al. (2006) [2]. For the large samples 18FAR02A and 18TIF01, a total of nine aliquots of three goethite bands were sampled, while 18 aliquots originate from three generations of the very large sample 18PL07. Each aliquot (0.02–0.08 mg) was measured, photographed, weighed and encapsulated into a 1 mm × 1 mm niobium tube (purity 99.9%). In a vacuum chamber, tubes containing goethite fragments were placed on a 49-cell plate and spaced out by 9 standard Durango apatite fragments packed on Pt tubes. Samples were degassed using a doped ytterbium laser under a high vacuum for 30 min at a temperature below 1000 °C in order to reach the thermal activation and diffusion of He without loss of U and Th by sublimation. The heating step was carried out once or twice, until full $^4$He degassing (signal down to the background level). The gas was subsequently expanded into the purification line and mixed with a known amount of $^3$He. In order to separate He from most $H_2O$, $CO_2$, $H_2$ and Ar, gas purification was performed by using three liquid nitrogen-cooled traps of activated charcoal, a titanium sponge trap heated at ~850 °C and a SAES 701 getter. Helium isotopes ($^3$He and $^4$He), $H_2O$, $CO_2$, $H_2$ and Ar were measured in the analysis section with a Pfeiffer Prisma Quadrupole mass spectrometer [7,30]. Aliquots were extracted from the vacuum chamber for chemical attack, and Nb tubes were transferred into 5 mL PFA capped vials (Savillex). Mineral digestion was carried out by adding 50 μL of 5N $HNO_3$ containing $^{235}$U (~4 ppb), $^{230}$Th (~4 ppb) and $^{149}$Sm (~4 ppb), 50 μL of 5N $HNO_3$, 400 μL of 40% concentrated HCl and two drops of 38% HF. The tightly closed vials were heated at about 100 °C overnight. The vials were then placed on a hot plate at 100 °C for two hours for complete evaporation. A 1.9 mL dose of 1N $HNO_3$ was added to the solution. The solution was heated at 100 °C to reflux for two hours. After cooling, 1.5 mL of the solution was taken and diluted with 1N $HNO_3$ to reach a total volume of 3 mL (Fe content <100 ppm). Finally, $^{238}$U, $^{232}$Th and $^{147}$Sm isotopes were determined using a high-resolution inductively coupled plasma mass spectrometer HR-ICP-MS (ELEMENT XR—Thermo Fisher Scientific). An analytical error of 5% at 1δ is expected for the two-step analysis based on Durango apatite dating uncertainties associated with the analysis made in parallel to the goethite. (U-Th)/He data are reported in Table S6.

## 4. Results

### 4.1. Goethite Characterization and Mineralogy

Botryoidal goethite in karst cavities and veinlets collected from the western part of the Imini district in Far West, Tifersine and Plateaux areas (Figure 3) show accretionary growth bands that indicate direct precipitation of successive generations of goethite (Figure 4). Goethite never replaces the dolostone fabric (Figure 5a,b) but is coated and intersected by later Mn oxides belonging to the coronadite group (Figure 5c). Botryoidal goethite consists of cohesive small crystals with homogeneous sections (Figure 5). The high level of homogeneity of the successive layers/generations of Imini botryoidal goethite is highlighted by SEM observations (Figure 5).

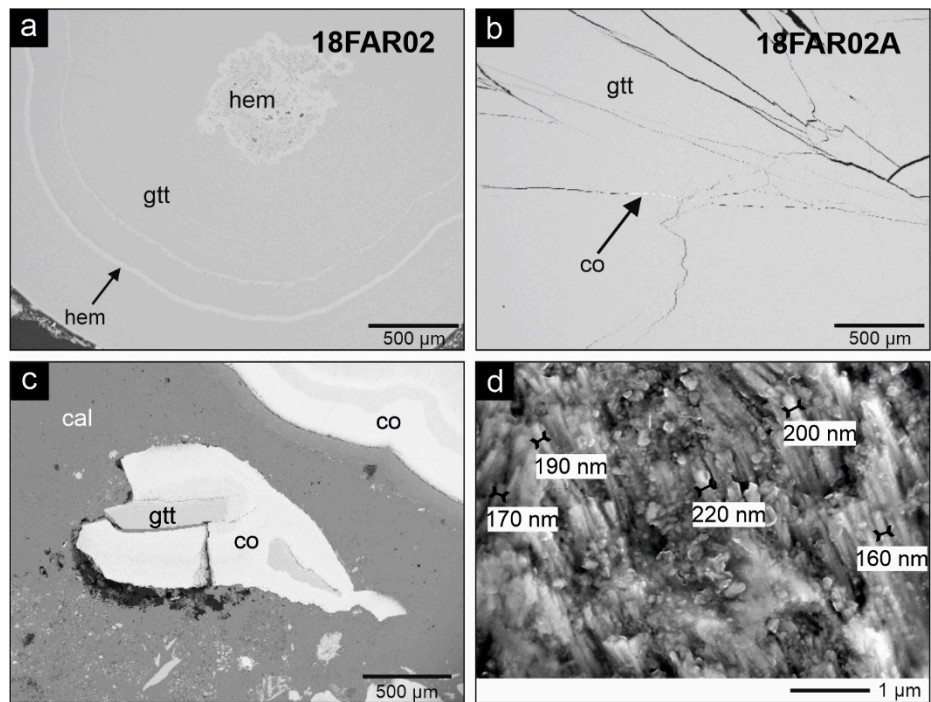

**Figure 5.** (**a–c**) SEM microphotographs of the most representative samples in back-scattered electron mode. (**d**) SEM microphotograph of botryoidal sample 18PL07. Secondary electron mode with the size of goethite crystallites (width exceeding 100 nm). Gtt = goethite; Hem = hematite; Co = coronadite group minerals; Cal = calcite.

The fluctuating colors of these layers (Figures 2 and 4b,c,e) are due to variations of porosity (Figure 4g). Scarce Mn oxides of the coronadite group occur in fractures (Figure 5b) and cavities of goethite, or as a coating on the last goethite generation (Figure 4g). These phases are never found between successive goethite layers. Hematite is occasionally observed at the contact between goethite and the dolomitic host rock, or the growth center of goethite (Figure 4a). Small, rare barite grains are noticed in sample 18TIF20. The width of goethite needles, which represents the smallest dimension, always exceed 100 nm, and the length is 1 to several µm long (Figure 5d).

*4.2. Goethite Geochemistry*

Imini goethite samples have homogeneous major, minor and trace element concentrations, as observed in Figure 6 and Tables S2–S4. They have moderate trace element concentrations, low rare earth element (REE) concentrations that do not exceed 20 ppm (sum REE) and similar REE patterns (Figure 6a,b). Most of the goethite has similar As content, but the variation of their Cu content that correlates with the U content is illustrated in Figure 6c,d. Cavity-filling samples present quite homogenous U content (~10 ppm; Figure 6d), whereas U content of the fracture-filling samples presents dispersed values (~1–15 ppm; Figure 6d). Sample 18TIF20 has distinguishing contents in Mn, Ba, As, Pb, Zn and Cu and higher REE concentration compared to other goethite (Figure 6). The enrichment in Ba is most probably related to the precipitation of barite or hollandite in cavities of more porous goethite layers.

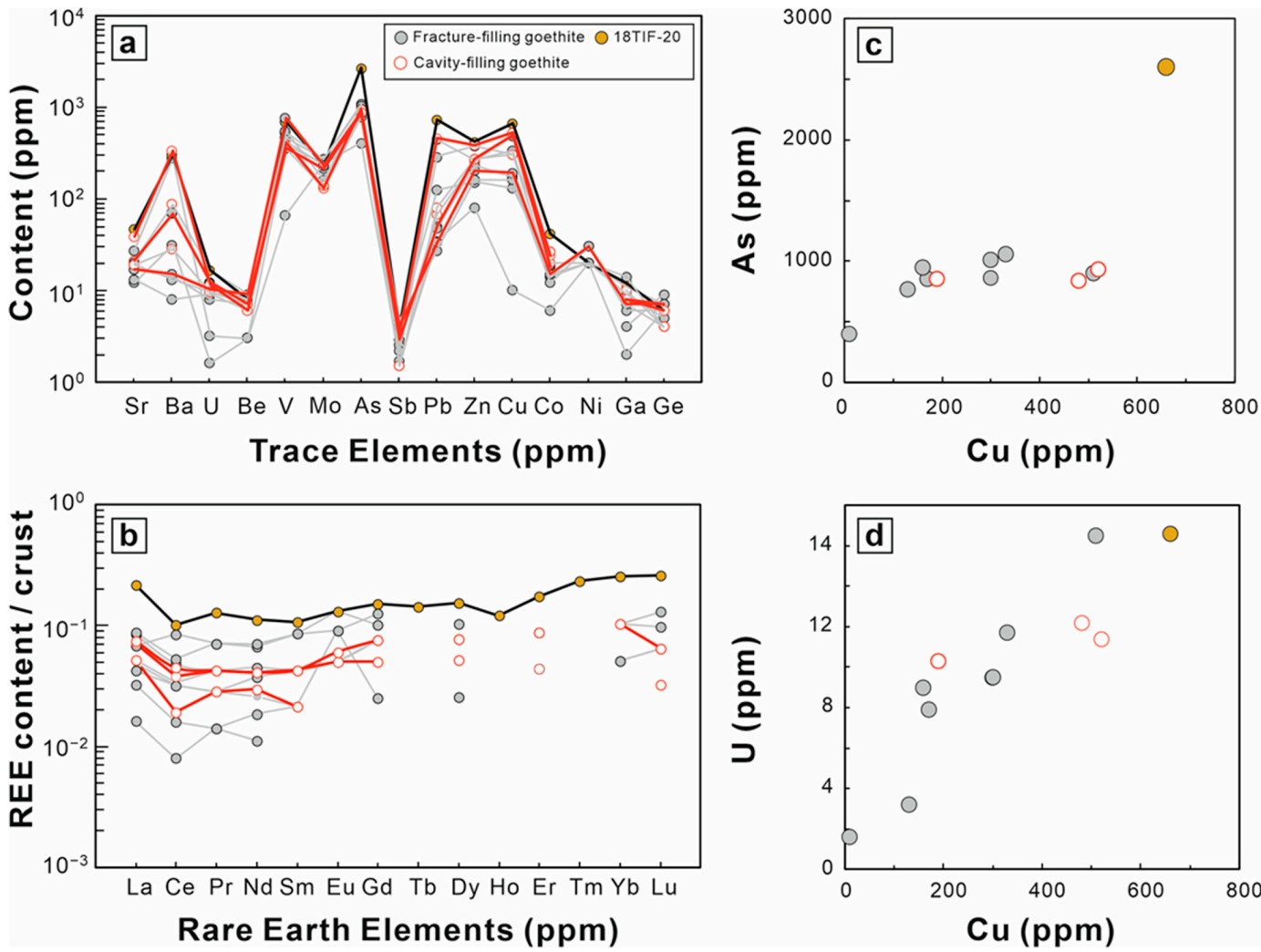

**Figure 6.** Geochemical data of Far West, Tifersine and Plateaux samples. (**a**) Trace element patterns; (**b**) rare earth element patterns normalized to the continental crust; (**c**) As vs. Cu contents and (**d**) U vs. Cu contents. The fracture-filling goethite is reported using gray circle symbols, except for 18TIF-20 where yellow-gray color is used, and the cavity-filling goethite is reported with a red open circle symbol.

### 4.3. Goethite $\delta D$ and $\delta^{18}O$ Values

The water content of Imini goethite has a range between 10.46 (18FAR02) and 11.41 wt.% (18PL02B) (Table S5). The VSMOW standardized $\delta D$ and $\delta^{18}O$ values range from $-172$ to $-148‰$ and 3.1 to 11.1‰, respectively, with different isotopic compositions between sample type (cavity and fracture-filling goethite) (Figure 7a). In addition, $\delta D$ values may differ, at equivalent $\delta^{18}O$ values, between cavity and fracture-filling goethite (Figure 7a).

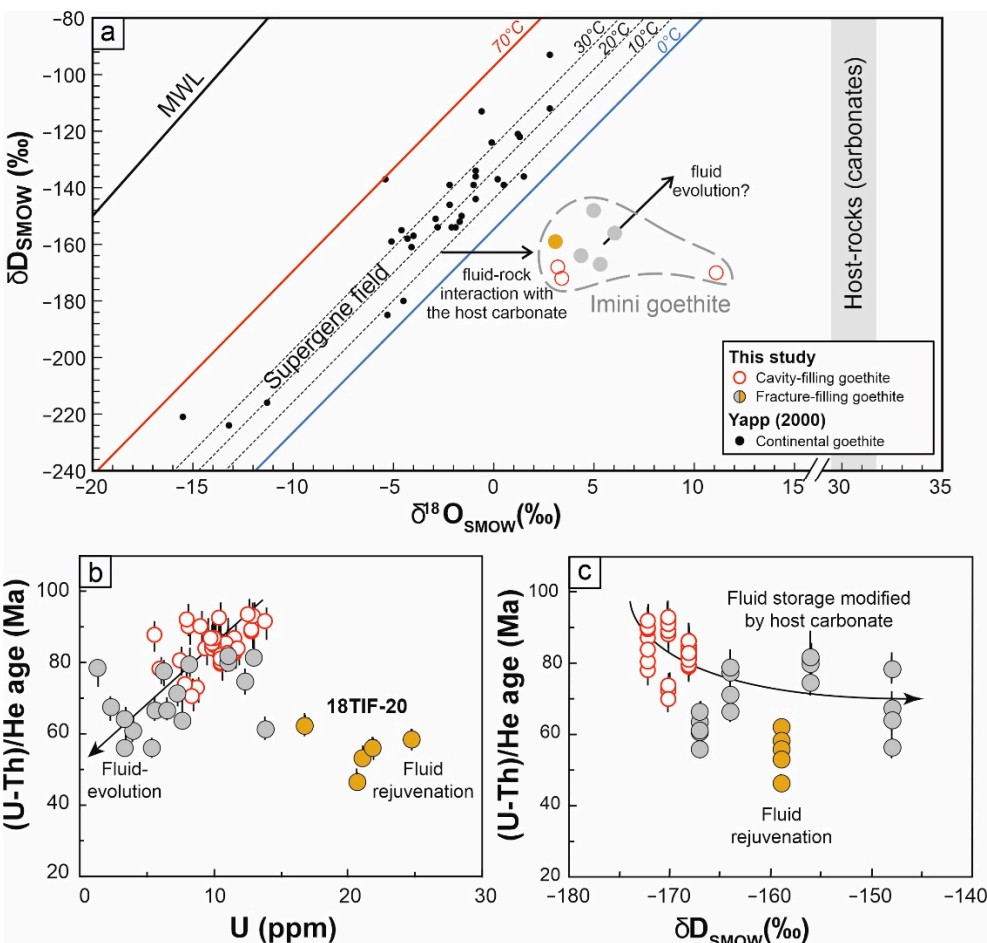

**Figure 7.** $\delta D$, $\delta^{18}O$ and (U-Th)/He data of Imini goethite. (**a**) Evolution of the $\delta D$ and $\delta^{18}O$ values of the studied goethite. Continental goethite from [31] is reported for comparison as part of the supergene field. Temperature lines are calculated from [32]. The meteoric water line (MWL) of [33] is given for reference. Carbonate ranges of values of Cenomanian-Turonian dolostones are given by the gray shade [34]. (**b**,**c**) Imini goethite GHe age variation as a function of the effective uranium U content and the $\delta D$ values, respectively. Cavity- and fracture-filling goethite (open red circle and orange circle, respectively) is reported. Sample 18TIF-20 fracture-filling goethite is distinguished from the other samples with a star filling the orange circle.

### 4.4. Goethite Age

Fifty-six goethite (U-Th)/He data were produced from eight samples; data are reported in Table S6. Three to six aliquots on three parts of the cavity-filling samples (18FAR02A, 18PL07, 18TIF01) (0.8 to 1.5 cm apart) (Figure 8) were selected, whereas the five fracture-filling goethite samples (18FAR02, 15PL02, 18PL02, 18PL25, 18TIF20) were dated without targeting particular generations. GHe ages have a range from $70 \pm 4$ to $93 \pm 5$ Ma for the cavity-filling goethite (Figures 8 and 9) and from $82 \pm 4$ to $47 \pm 2$ Ma for the fracture-filling goethite (Figure 8a,b), respectively.

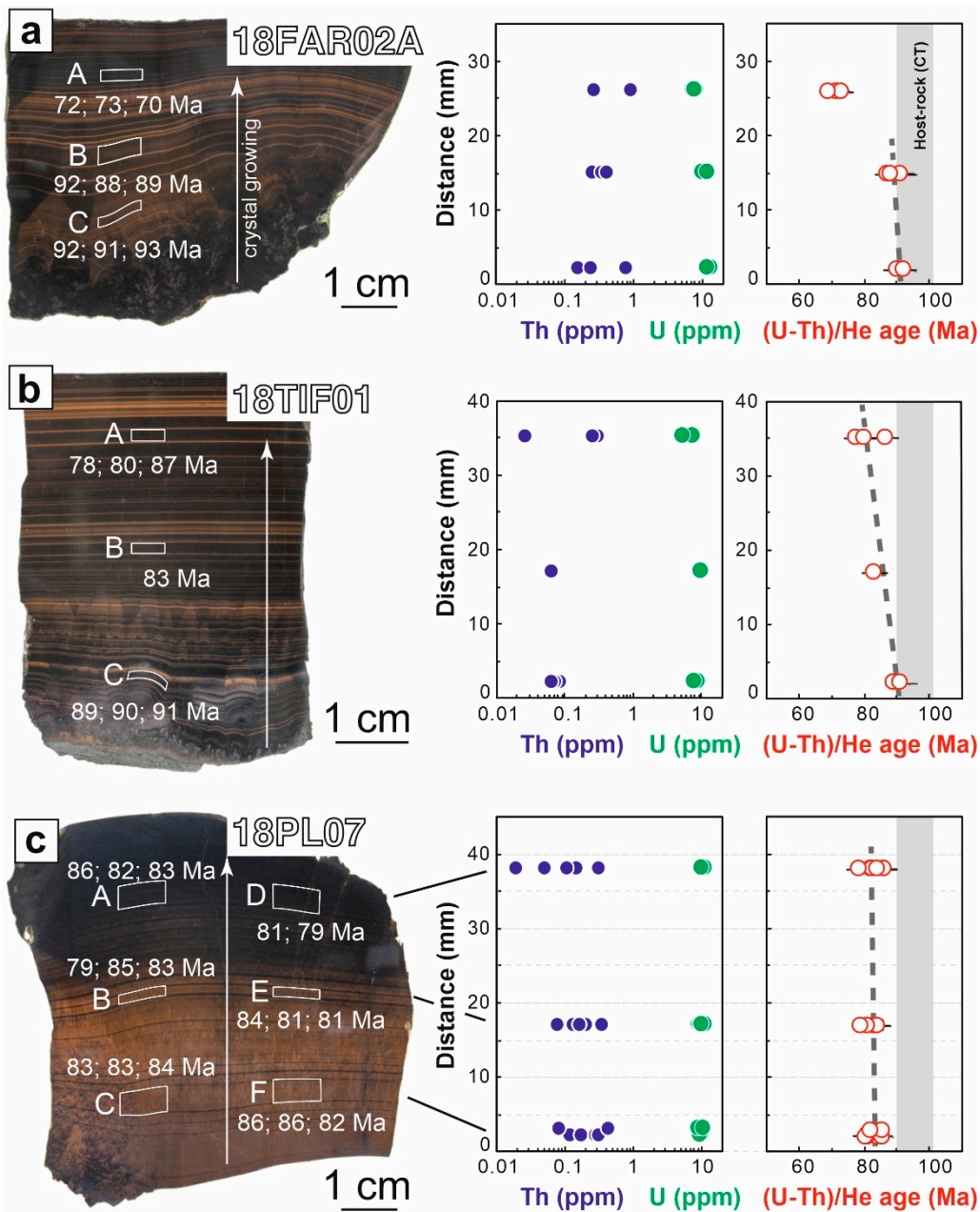

**Figure 8.** Cavity-filling botryoidal goethite samples (**a**) 18FAR02A, (**b**) 18TIF01 and (**c**) 18PL07 and evolution of Th and U contents and (U-Th)/He ages along transects of the samples. On the sample pictures, the letters A to F and the white rectangular zone refer to the sampling zone given in Table S6.

The particularly reproducible (U-Th)/He ages of the present study are characterized by a very low dispersion: 4% for goethite rather than >6% as generally found for (U-Th)/He methods with other minerals such as apatite or zircon [30,35]. The error associated with the He diffusive losses is estimated to represent a maximum of 5% of the ages, including analytical errors. This value is much lower than those usually estimated in previous studies [3,7–10]. The high reproducibility of our ages arises contingently from the immediate precipitation of goethite in fractures and cavities, the well-arranged botryoidal texture (Figures 8 and 9), the nearby absence of inclusions and contaminant minerals, the rather large dimensions of growth crystallites that ensure an efficient He retention (lowest width > 100 nm; Figure 5d), the very low content in elements (Figure 6) suggesting only few substitutions and, consequently, few intrusions in the crystal lattice. Compared to

previous studies, the rare combination of all these parameters in goethite most likely led to very limited He diffusive losses (e.g., [3,6,10]).

All GHe ages are younger within the error margins than the Cenomanian-Turonian host rock (100.5–89.8 Ma) and imply that GHe required very few (<2%) or even no He loss correction [2,3]. This is consistent with the Imini goethite's large crystallite size (width > 100 nm and length > 1 μm; Figure 5d) that enables a remarkably high He retention (>98%). Only GHe ages without He loss correction with ±5% error are discussed here. Goethite's effective uranium U content ([U] + 0.238 × [Th] + 0.0012 × [Sm]) [36] is carried mainly by the U content as the Th and Sm contents are very low in comparison (Table S6). The U content is homogenous (around 10 ppm) in cavity-filling goethite, showing slightly lower content for the youngest goethite layers (Figure 7). Fracture-filling goethite has the same behavior with younger ages and lower U content (Figure 7b). Only the 18TIF-20 sample, which presents the youngest (U-Th)/He ages, has a higher U content (15–25 ppm) compared to other cavity and fracture-filling goethite (Figure 7b). The geochemistry of this sample also deviates from the other samples, with higher REE content; the highest Mn, Ba, Cu, As, Pb and Zn; the lowest Fe and V and distinct MnO, CaO and As concentrations (Figure 6).

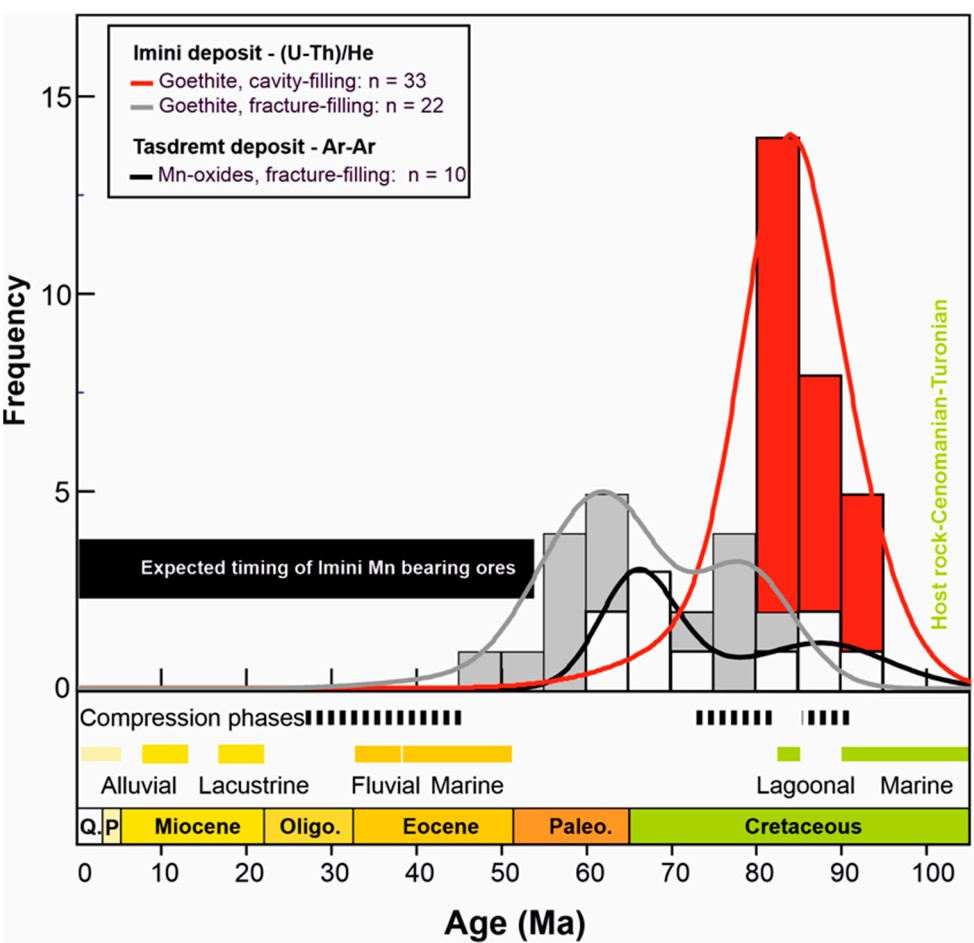

**Figure 9.** Imini cavity and fracture-filling goethite (U-Th)/He (this study) and Tasdremt coronadite group minerals $^{40}$Ar/$^{39}$Ar [14]: histogram of age distribution. The late Cretaceous and middle to early Cenozoic compressional phases [15,37] and temporal information of sedimentological deposits [16,23] are also reported.

## 5. Discussion

### 5.1. Meteoric-Derived Mineralizing Fluid at the Origin of Goethite

The combined δD-δ$^{18}$O values, geochemical composition and GHe data highlight the evolution of the mineralizing fluid composition as early as 95–90 to 50 Ma, corresponding

to the first mineralizing fluid circulation that has allowed the concentration of metals in the goethite parent fluid. The formation process of Mn oxides in the karst-hosted Mn deposits of Imini-Tasdremt involves a joint increase in pH and Eh [12–14] due to dolomite dissolution and the oxygenated environment. This feature strictly differs from weathering crust formation [38–40], where oxidation prevails. Goethite precipitation would have occurred under similar conditions, and its position prior to the Mn oxides in the paragenetic sequence (Figure 2) is explained by a stability field at lower Eh-pH than the Mn oxides [41].

Imini goethite δD-δ$^{18}$O values cluster below the 0 °C temperature or fractionation line of [32] (Figure 7a; Table S5), which means that goethite did not precipitate in equilibrium with surficial meteoric waters. A difference in altitude during the ~40 Myr of goethite precipitation would impact the δD and δ$^{18}$O values of surficial meteoric waters [42]. However, the late Cretaceous elevation of the Imini area did not exceed a few meters a.s.l. [23], which limits variations in the δD-δ$^{18}$O values of the mineralizing fluid. The change toward negative values of the δD is noted for fracture-filling goethite compared to cavity-filling goethite (Figure 7a,c). Hence, the goethite isotopic composition depends on its (i) formation process and (ii) timing. We cannot discriminate the importance of these two items, but it is obvious that the mineralizing fluid composition evolved through time. A change in the source, temperature or mixing with other fluids can be invoked, in the context of tectonic instability [15] and repeated inwards of seawater in the area between exposure events [23]. The important offset of isotopic values to the right of the supergene field (Figure 7a) can be related to fluid–rock interaction with the carbonate host rock. Dekoninck et al. [43] have shown that $^{18}$O-rich kaolinite-halloysite in the Tamra Fe(-Mn) deposit of Northern Tunisia results from heated meteoric waters that have exchanged their oxygen with the carbonate basement (marls). At Imini, the host rock dolostone has high δ$^{18}$O values (28.7‰–31.8‰) [34] and could exchange its heavy oxygen with the mineralizing fluid and precipitate $^{18}$O-rich goethite. The subsurface context of karstification, mineralogical sequence and deposition of Mn and Fe ores in the Imini area suggests that mineralizing fluids have a meteoric origin. The magnitude of the change in the δ$^{18}$O values of the fluid by isotopic exchange reactions between water and dolostone is calculated at different temperatures for different water/rock ratios using the Equation (1) of [44]:

$$\delta^{18}O_w^f = \frac{\delta^{18}O_{dol}^i - \Delta\delta^{18}O_{dol-w}^f + \left(\frac{w}{r}\right) \cdot \delta^{18}O_w^i}{1 + \left(\frac{w}{r}\right)} \tag{1}$$

in which $\delta^{18}O_{dol}^i$ is the initial oxygen isotope composition of the host rock dolostone (31.8‰) [34]; $\Delta\delta^{18}O_{dol-w}^f$ is the fractionation factor between dolomite and water from [45] ($\Delta\delta^{18}O_{dol-w}^f = 10^3\ln(\alpha) = 18.4$ and 31.8 at 100 and 25 °C, respectively) and $\left(\frac{w}{r}\right)$ is the molar ratio of exchangeable oxygen atoms in the water to those in the rock. $\delta^{18}O_w^i$ is the initial δ$^{18}$O value of water estimated at −10‰ and is a safe estimation of the original δ$^{18}$O composition based on δD values and the global meteoric water line (MWL). We assume that hydrogen did not exchange with the carbonate host rock as it contains no hydrogen. From this isotopically modified water ($\delta^{18}O_w^f$), due to fluid–rock interaction, the theoretical composition of equilibrated goethite was recalculated using the equation of [46] for meteoric fluids. The results show that fluid–rock interaction at 25 °C cannot produce $^{18}$O-rich goethite from isotopically modified water, even at low water/rock ratios (<0.1). Only temperatures above 70 °C can produce such $^{18}$O-rich goethite at a low water/rock ratio (<0.1 at 70 °C and <0.4 at 100 °C). Since we have no other compelling arguments supporting a hot $^{18}$O-rich meteoric fluid, it is likely that goethite precipitated from a fluid that is not completely meteoric. We lack data to determine its origin, such as late Cretaceous surface temperatures, the initial δ$^{18}$O values of the fluid and the exact water/rock ratio. Such high temperatures cannot be reached by burial of the sedimentary pile (~160–220 m) [23,47], i.e., connate waters. We cannot rule out a hydrothermal input connected to the late Cretaceous

doming [15,48] since compressional events would facilitate migration of hot brines along fractures in the basin. This event is difficult to constrain given its low intensity and that the three steps of the Atlas building obliterate most of the late Cretaceous structural record [16]. There are also no vein-type deposits of the same age in the area or hydrothermal pipes directly connected to the Imini mineralization. The contribution of sea water is unlikely given that the Imini area was rarely flooded during the late Cretaceous (Figure 9) [23]. A mixing fluid model, as supported by [13,14], seems more relevant and fits with the observed isotope compositions: metal-rich and anoxic mix with oxygen-rich surficial water resident in the open karst system. The presence of evaporites in the late Cretaceous series (Figure 1c) would also affect the composition of the initial fluid, but this would change both the hydrogen and the oxygen isotope compositions. Moreover, the equation from [46] is used for pure meteoric water to recover the $\delta^{18}O$ values of goethite. Besides these uncertainties, stable isotope compositions of goethite confirm that the Imini mineralizing system works differently compared to supergene fluids circulating in weathering crust. The Imini karst system shows the dualistic feature of Mn ore karst deposits with mixing of surficial meteoric waters and likely deeper basinal solutions [49]. These deposits occur in slightly dislocated and unmetamorphosed basins containing accumulation of ground waters over discontinuous and long periods. The Imini deposit remained close to the surface during its deposition and is different from known karst-hosted deposits worldwide where burial is at the depth of 4–6 km and temperatures of the fluids in the range 100–200 °C [49,50].

*5.2. Timing of Goethite Precipitation*

The high-spatial resolution dating of cavity-filling goethite (18FAR02A, 18TIF01 and 18PL07) provides similar GHe ages from 95 to 85 Ma for the first generation (Figure 9). The decrease in GHe ages from the border to the core confirms that ages decline as a function of goethite growth and testifies for the evolution of the mineralizing fluid (Figure 8). Change in GHe age populations of sample 18FAR02A reveals the rapid precipitation of goethite around 90 Ma of a centimeter of goethite and the precipitation of thinner layers around 70 Ma (Figure 8a). A short precipitation period is also suggested for the similarly aged thick layers of sample 18PL07. Finally, the change in the geochemistry shown in sample 18-TIF-20 (Figures 7 and 8) indicates the arrival of a different parent fluid at ~60 Ma. The fluid is still $O_2$-poor and acidic in $Fe^{2+}$ and $Mn^{2+}$ mobilization, but the temperature, the trajectory or the source may have changed. Goethite precipitation could have occurred continuously or in a stepwise manner during a period of around 40 Myr (Figure 9). However, the number of GHe ages obtained in the Imini district cannot discriminate one scenario from the other. Nevertheless, a continuous goethite precipitation over 40 Myr is difficult to justify from a hydrological and geological point of view, as it implies the same conditions over that period. In addition, the $^{40}Ar/^{39}Ar$ dating of coronadite group minerals in the co-genetic Tasdremt deposit [14] supports a stepwise scenario. For this reason, we further interpret from these botryoidal samples that the Fe mineralization occurred in a stepwise manner and not continuously over a period of about 40 Myr (Figure 9). GHe ages of cavity-filling goethite suggest the precipitation shortly after the dolostone formation and repeated inputs of mineralizing fluids leading to goethite precipitation. In addition, the GHe ages obtained for the fracture-filling goethite samples (18FAR02, 15PL02, 18PL02, 18PL25, 18TIF20) show that precipitation started around 80 Ma (Figure 9), only 10 Ma after the cavity-filling goethite, and continued up to about 50 Ma.

Finally, the changes in $\delta D$ values, GHe ages and U content of the parent-fluid composition through time (Figure 7b,c) likely involved tectonics as a controlling factor of the fluid composition. Tectonic activity clearly appears after the Santonian (Figure 9), when the mineralization system opened with fracture-filling mineralization. Accordingly, cavity-filling goethite precipitated with similar $\delta D$ values but variable $\delta^{18}O$ (Figure 7a,c) and homogeneous U content (Figures 7b and 8) from 95 to 80 Ma. From an initial parent fluid, the composition evolved since fracture-filling goethite appeared at about 80 Ma (Figure 7b). Such correlation between textural change and fluid composition of the Fe oxide supports

an opening of the system and the input of fluids having a slightly different composition. Around 65 Ma, a new parent fluid precipitated goethite 18TIF-20 that shows systematically higher U content (Figure 7b) and has a distinct geochemical composition. This last event appears in the same time window as the last Mn mineralization event observed by [14].

### 5.3. Formation Model and Regional Implication

The meteoric-derived mineralizing fluid reacted with the host-carbonate and precipitated a first generation of goethite as cavity-filling and subsequently a second generation as fracture-filling goethite (Figure 9). A compressional regime would be responsible for the first occurrence of fracture-filling goethite, with an age of about 80 Ma that might reflect early Atlas tectonic activity (Figure 9). This is consistent with the geodynamic activity identified in the area associated with global doming of the High Atlas (e.g., [15,37]), especially as the host rock dolostone remained above sea level since the early Turonian, except during the upper Santonian [23]: most of the goethite precipitated under a cover of hundreds of meters of sediments after the upper Santonian [23,47]. The surface uplift initiated by the late Cretaceous High Atlas doming [15] would be responsible for repeated inputs of subsurface meteoric-derived mineralizing fluids. The depth of Fe and Mn oxide formation would correspond to the thickness of the overlying Senonian sedimentary pile but during episodes of global exposure of the land surface that correlate with the late Cretaceous doming.

The GHe ages, coupled to macroscopic and microscopic observations, also indicate crystallization of goethite before the Mn oxides that crosscut goethite (Figures 2, 3 and 5). Furthermore, the GHe geochronology is consistent with the $^{40}$Ar/$^{39}$Ar ages obtained in the nearby Tasdremt Mn ore (Figure 1b) [14]. At Tasdremt, coronadite group minerals are contemporaneous with the fracture-filling goethite mineralization, which may suggest a similar genesis between the deposits. Although crosscutting relations exist between some Fe and Mn oxides (Figures 2 and 5c), their similar age distribution (with different accuracy) suggests they occur during the same mineralization pulses but in very different proportions. Accordingly, Mn oxides are predominant, when Fe oxides are localized. The reason why Fe is perfectly separated from Mn is likely its sequential precipitation in environments that become increasingly oxidizing with time [51]. However, at Imini, some of these K-Mn oxides and most of the pyrolusite also crosscut both the cavity and fracture-filling goethite, implying these Mn oxides are younger (Figure 2). According to e.g., [13,14], the doming of the Imini-Tasdremt area during the late Turonian allowed exposure, partial erosion of the uppermost carbonates and development of a karstic surface at Imini. The concentration of botryoidal goethite exclusively in the upper part of the Cenomanian-Turonian mineralized level is probably also related to this early dissolution event.

## 6. Conclusions

This study emphasizes that goethite precipitated under particular subsurface conditions in the Imini Mn ore deposit. The $\delta$D and $\delta^{18}$O values of goethite show that the mineralizing fluid was enriched in $^{18}$O, most likely due to fluid–rock interaction with the host rock dolostone during carbonate dissolution or mixing fluids. These conditions likely favored well-crystallized goethite, allowing the access to highly reproducible (U-Th)/He ages and the identification of a goethite precipitation regime with time. The cavity-filling goethite precipitated between 95 to 80 Ma, whereas fracture-filling goethite formed between 80 to 50 Ma. The progressive increase in $\delta$D values and the depletion in U content attest to a change in the mineralizing fluid composition. The final Fe mineralization stage at ~50 Ma is notably enriched in U, Cu and trace metals. These changes are most likely linked to the tectonic instability regime and changes in the source, trajectory or temperature of the mineralizing fluid. It is unlikely that the ore system was active over the entire 40 Myr period but was rather (re)activated several times according to orogenic pulses related to the early Atlas doming.

**Supplementary Materials:** The following supporting information can be downloaded at: https://www.mdpi.com/article/10.3390/min12091151/s1, Table S1: Summary of information and performed analyses of Imini goethite, Table S2: Results of geochemical analyses for major elements, Table S3: Results of geochemical analyses for rare earth elements, Table S4: Results of geochemical analyses for minor elements, Table S5: Oxygen and hydrogen isotopic compositions of Imini goethite, Table S6: Imini goethite (U-Th)/He data.

**Author Contributions:** Conceptualization, M.V., C.G., A.D. and J.Y.; methodology, M.V., R.P.-J., C.G.; validation, M.V., C.G. and J.Y.; formal analysis, R.P.-J., M.V. and T.V.; investigation, M.V. and A.D.; writing—original draft preparation, M.V., C.G., A.D. and J.Y.; writing—review and editing M.V., C.G., A.D., T.V., R.P.-J., A.M., J.Y; visualization, M.V., C.G. and A.D.; supervision, C.G. and J.Y.; project administration, C.G. and J.Y.; funding acquisition, M.V., J.Y. and C.G. All authors have read and agreed to the published version of the manuscript.

**Funding:** This research received funds from the FRS-FNRS through a FRIA PhD grant. The GHe analysis was funded by the RECA ANR-17-CE01-0012-01 project.

**Data Availability Statement:** Not applicable.

**Acknowledgments:** M. Verhaert thanks the Belgian National Fund for Scientific Research, for providing a FRIA PhD grant. We thank A. Maali, A. Ait Bendra and K. Ikken from the SACEM for the field access, M. Essalhi (University Moulay Ismaïl, Errachidia) for his help in the field, F. Haurine (University Paris-Saclay) for the ICP-MS analyses, G. Rochez (University of Namur) for the XRD analyses and E. Pirard and H. Bouzahzah (Liège University) for the use of the SEM. This research used resources of the Electron Microscopy Service of the University of Namur. We are thankful to two anonymous reviewers for improving the manuscript with relevant comments about the content.

**Conflicts of Interest:** The authors declare no conflict of interest. The funders had no role in the design of the study; in the collection, analyses, or interpretation of data; in the writing of the manuscript or in the decision to publish the results.

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
