# Peer review of "Unravelling the Temporal and Chemical Evolution of a Mineralizing Fluid in Karst-Hosted Deposits: A Record from Goethite in the High Atlas Foreland (Morocco)"

_minerals, doi:10.3390/min12091151_

Round 1

Reviewer 1 Report

In the manuscript ID: minerals-1876307, the authors studied the geochemical evolution and dating of several karst-hosted deposits in the High Atlas Foreland (Morocco). This paper is interesting because it provides new data and insights into the progressive evolution of mineralising fluids through the analysis of goethite crystals. The authors present quality data and content suitable for publication in Minerals. However, the manuscript requires some modifications before being published.

 I congratulate the authors, who have done a great job and produced a high-quality paper.

 Most comments below are not criticisms to be addressed absolutely but mere suggestions for improvement, and I trust the authors to know better than I which ones are valuable to follow.

COMMENTS

 General

 A valuable paper. The paper is competently written and easy to read and has been performed using appropriate techniques to arrive at the conclusions provided by the authors. The writing is clear, but parts need improvement, and the structure is satisfying. Syntactic and minor grammatical errors require a thorough review by the authors.

 Using XRD, SEM, FUS-ICP, stable isotope geochemistry and (U-Th)/He dating, the authors address the age and geochemistry of the goethites. However, some general data on the mineralogy and description of the deposits (tonnage, cut-off…) would improve the paper.

 They synthesise information obtained as pictures and diagrams.

Main Comments

 My main comments on the paper are:

1) The paper is well structured. However, some parts need to be improved, especially the discussion.

2) Information about the resources of the deposits would be very welcome.

3) The data collection methodology is adequate for the work in question.

4) Figures are sufficient and perfectly support the authors' arguments. The design of the figures is beautiful and presents very clear information.

5) Some phrases must be rewritten to provide more clearness to their text.

6) The authors argue that the fluids are essentially meteoric, and all the modelling calculations are along those lines. Indeed their arguments are convincing, but I think it would improve their argument if they show evidence that discards the intervention of seawater or connate water or the presence of a hydrothermal source related to doming. For example, Frizon de Lamotte et al. (2009) suggest a thermal doming cause for the uplift based on the magmatic activity in the Central High Atlas during the late Mid Jurassic-Early Cretaceous. Therefore, would the possibility of hydrothermal fluids in the context of these mineralisations be unfeasible?

7) Furthermore, Figure 9 shows that during the Cretaceous and Paleocene the region was in a marine environment, so the authors should clarify how if the basin waters are marine, only meteoric waters are involved in the mineralisation.

 Although all comments are included in the attached document, here I highlight some that could be taken into consideration:

·         In the sketch of figure 3A, the arrow of the dolomite points to the Mn veins. Please change the position of the arrow a little.

·         Line 429: "Mn oxides are economic, when Fe oxides are not." You can argue the presence of different concentrations but not directly say that some are economic only because of concentration, since other aspects govern the economic viability of a deposit, essentially by the cut-off grade where the value of the metal to be mined is of essential importance.

 All other comments are included in the attached document. Excuse me for including several files; I had a problem with the software to edit the pdf files.

References

The bibliography has been well used, including the most relevant papers in the field. The authors don't follow the Minerals standards for reference because they do not use the journal's abbreviations.

Typos and nitpicking

Typographical errors and nitpicking are included in the attachment. I also have some suggestions about the text.

Author Response

Dear reviewer,

We are very grateful for the relevant comments about the manuscript and the careful editing of the English. This is much appreciated. We are confident that the review is of quality and the questions you addressed significantly increase the quality of the manuscript. We have considered all the comments and we have changed the manuscript to fit with these comments . We accept all changes in the annotated pdf files. The reply to the major comments are given below:

  • We have carefully added some elements in the discussion to make the message clearer.
  • We add information about resources and grades in the geological setting.
  •  
  • We have added some information in Figures 3, 5, 7 and 8 as requested.
  • We rewrote some sentences.
  • We enlarge the discussion to integrate other type of fluids, especially the hydrothermal contribution. See section 5.1.
  • This is a very interesting comment. There is indeed a short sedimentary pulse during the Santonian which deposited almost 150 m of “Senonian” sediments over the Cenomanian-Turonian. In addition the Imini deposit as long been considered as a sedimentary deposit for mineral exploration (this is the current model the operator follows). Actually, this is rather dolomitization that makes conditions suitable for Mn oxide formation. This sedimentary model has been ruled out by several papers since 2006 (see list of refs in the text). So, it is unlikely that sea water is involved. Nevertheless, you are right that this must be mentioned and we did so in section 5.1. Best regards.

Many thanks.

Best regards.

Reviewer 2 Report

Dear authors,

I have reviewed the manuscript entittled "Unravelling the temporal and chemical evolution of a mineralizing fluid in karst-hosted deposits: a record from goethite in the High Atlas foreland (Morocco)"

The paper is very well written and the method is of great interest in order to date processes of fluid migration. However there are several points that need to be addressed before the publication of this work. Most of them are minor and are includded in the annotated text. Despite this, my major concern is that this study is very local. I would like to see in the discussion a comparison with other similar karst-hosted deposits. Also, It would be interesting to provide fracture orientation data in order to correlate the mineralizations with the recional tectonics. Finally, some of the points discussed are vague and and should be extended (e.g. line 332).

I know that the authors will not have problems in order to fix these issues. For that reason I recomment moderate revisions before the acceptance and publication of the manuscript.

Sincerely

Author Response

Dear reviewer,

We are very grateful for the relevant comments all along the manuscript. We are convinced that the review is of quality and the questions you addressed significantly increase the quality of the manuscript. We have considered almost all the comments and we have changed the manuscript to fit with the comments. These items are directly put as “reply” in the annotated pdf. One can see the changes in the track-changes file.

The reviewer misses a global implication of our work and the comparison with other similar deposits. We actually lack some similar deposits in other countries. This is why our team have published so much about the district over the last 10 years. We try to understand how possible it is to develop our knowledge of this type of mineralization and which conditions differ from strictly supergene deposits. Actually, there is one similar deposit in Tunisia but it has not been studied in details yet (Garnit et al. 2021). In the world, karst-hosted deposits occur at higher depth with other mineralogy. We mention them in distinction to enlarge the topic and fit with this recommendation at the end of section 5.1. Still, the sentence highlights such difference and we explained in section 5.1 the parameters of the fluids based on our dataset. This is a global implication of our work, which demonstrates that goethite may have a different composition than that in weathering crusts.

About the fracture data, we did not measure them in the field. So, we are not able to provide them.

We carefully reply to the other recommendations.

Best regards,

Round 2

Reviewer 1 Report

The authors have improved the previously submitted paper. In addition, they have clarified many points that the reviewer might not have understood well. They had little to improve, as they had done excellent work previously.

Thank you very much for the effort made and for your reply. This reply clarified your approach and is very well justified and argued. I consider that the paper meets Minerals' quality parameters for publication.

I only have a suggestion about the reference list since you have not abbreviated all the journals. For example, Economic Geology = Econ. Geol. or Mineralium Deposita = Miner. Depos. Please check the abbreviations.

Congratulations on your work, and thank you for viewing some of my suggestions.

Best regards,

Reviewer 2 Report

The authors have been addressed most of my suggestions and have provided proper answers to my questions. Therefore, the manuscript can be accepted for publication.

Best regards